# Towards Modality-Agnostic Continual Domain-Incremental Brain Lesion Segmentation

**Yousef Sadegheih**[1]  iD                                      Yousef.Sadegheih@ur.de
**Dorit Merhof**[1,2]  iD                                        Dorit.Merhof@ur.de
**Pratibha Kumari**[1]  iD                                       Pratibha.Kumari@ur.de

[1] *University of Regensburg, Regensburg, Germany*

[2] *Fraunhofer Institute for Digital Medicine MEVIS, Bremen, Germany*

**Editors:** Accepted for publication at MIDL 2026

## Abstract

Brain lesion segmentation from multi-modal MRI often assumes fixed modality sets or pre-defined pathologies, making existing models difficult to adapt across cohorts and imaging protocols. Continual learning (CL) offers a natural solution but current approaches either impose a maximum modality configuration or suffer from severe forgetting in buffer-free settings. We introduce **CLMU-Net**, a replay-based CL framework for 3D brain lesion segmentation that supports *arbitrary and variable* modality combinations without requiring prior knowledge of the maximum set. A conceptually simple yet effective channel-inflation strategy maps any modality subset into a unified multi-channel representation, enabling a single model to operate across diverse datasets. To enrich inherently local 3D patch features, we incorporate lightweight domain-conditioned textual embeddings that provide global modality-disease context for each training case. Forgetting is further reduced through principled replay using a compact buffer composed of both prototypical and challenging samples. Experiments on five heterogeneous MRI brain datasets demonstrate that CLMU-Net consistently outperforms popular CL baselines. Notably, our method yields an average Dice score improvement of $\geq 18\%$ while remaining robust under heterogeneous-modality conditions. These findings underscore the value of flexible modality handling, targeted replay, and global contextual cues for continual medical image segmentation. Our implementation is available at https://github.com/xmindflow/CLMU-Net.

**Keywords:** Continual learning, Arbitrary brain MRI modality, Brain lesion segmentation

## 1. Introduction

Magnetic Resonance Imaging (MRI)-based brain lesion segmentation is vital for diagnosis, treatment planning, and longitudinal monitoring of neurological disorders (Despotović et al., 2015; Sadegheih et al., 2025a). However, clinical deployment faces substantial challenges due to the dynamic nature of clinical data (Kumari et al., 2024, 2025a,c). Modality availability, acquisition protocols, and pathology distributions vary widely, and new imaging sequences or auxiliary information continue to emerge. Different hospitals and research centers produces cohorts with diverse modality configurations, pathology, and patient populations, yielding highly heterogeneous MRI datasets. Conventional U-Net based models are typically trained for specific modality-pathology combinations, necessitating retraining or separate cohort-specific models, a practice that is resource-intensive and restricts generalization. Alternatively, joint training approaches aim to learn a single model that handles

variable modality configurations (Xu et al., 2024; Zhang et al., 2025), but they require simultaneous access to all datasets, which is rarely feasible due to privacy, acquisition timing, and storage constraints. These models also degrade when test cohorts exhibit modality or pathology distributions not seen during training, often necessitating retraining.

A more realistic setting is to learn the datasets sequentially as they become available. Naively updating a U-Net in this scenario leads to Catastrophic Forgetting (CF), where performance on earlier datasets deteriorates sharply (Chen and Liu, 2022). Continual Learning (CL) offers a framework to alleviate this problem by enabling models to integrate new information while retaining prior knowledge (Kumari et al., 2025b). However, CL methods developed for natural images often underperform in medical segmentation, where dense prediction significantly amplifies forgetting (González et al., 2023). Lifelong U-Net (González et al., 2023) demonstrated that buffer-free approaches struggle to maintain segmentation accuracy and perform substantially worse than joint or cumulative training. Rehearsal-based methods, which replay a small subset of past samples, offer stronger retention but still achieve limited backward transfer. While experience replay is highly effective in natural image and audio domains (Rolnick et al., 2019; Bhatt et al., 2024), its exploration under heterogeneous-modality MRI settings remains limited. Beyond mitigating forgetting, continual brain MRI segmentation must also accommodate variable modality across datasets. Different cohorts are acquired with differing sets and counts of modalities, and future acquisitions may introduce modalities unseen in earlier stages. A recent modality-agnostic framework (Sadegheih et al., 2025b) demonstrated the feasibility of continual segmentation under heterogeneous modality inputs, yet CF remained substantial. Moreover, the framework assumes prior knowledge of the maximum number of modalities across all datasets, which prevents seamless extension to cohorts containing novel or auxiliary sequences. As imaging protocols continue to evolve, such constraints limit clinical applicability. A continual segmentation system must therefore support arbitrary and evolving modality combinations while maintaining robust performance across sequential domains.

We propose the Continual Learning in Modality-agnostic U-Net (CLMU-Net), a replay-based framework designed for brain lesion segmentation under dynamic and heterogeneous MRI conditions. CLMU-Net integrates a lesion-aware replay buffer with lightweight textual conditioning that encodes concise descriptions of modality availability and lesion characteristics. Cross-attention injects these domain-aware cues into bottleneck features, guiding the network toward domain-appropriate representations. A conceptually simple yet effective channel-inflation mechanism enables arbitrary modality subsets without requiring a predefined maximum set, allowing seamless adaptation as new cohorts or modalities appear. Together, these components allow CLMU-Net to learn sequentially with substantially reduced forgetting. We evaluate CLMU-Net on five diverse brain MRI datasets spanning different lesion types, modality configurations, and acquisition centers. Across two dataset-order permutations, CLMU-Net consistently outperforms both buffer-free and rehearsal-based baselines, yielding higher Dice scores and improved stability under variable-modality conditions. Our key contributions are as follows: ❶ We introduce a lesion-aware replay buffer that prioritizes structurally informative and uncertain samples, improving knowledge retention under strict buffer budgets. ❷ We develop a domain-conditioned textual guidance mechanism that injects global modality and lesion cues into bottleneck features through cross-attention. ❸ We propose a modality-flexible input mechanism based on chan-

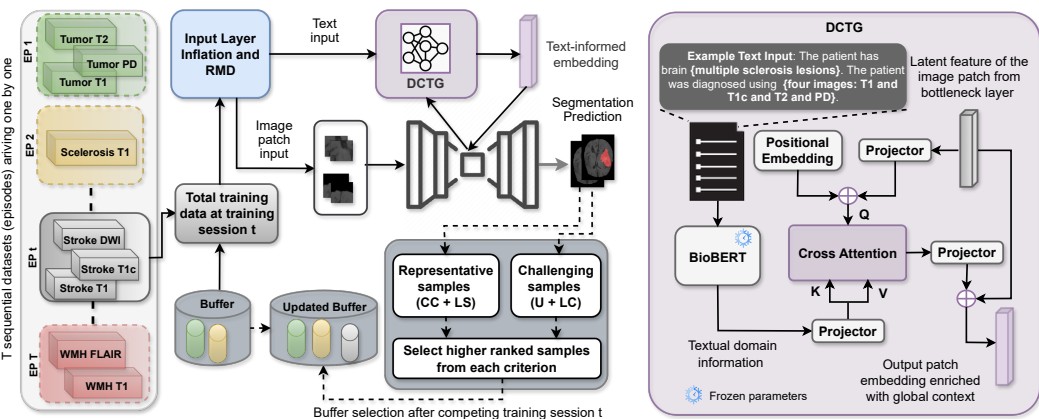

Figure 1: Overview of the CLMU-Net framework.

nel inflation that supports arbitrary and previously unseen modality combinations without predefined limits. ❹ We demonstrate consistent performance gains across five heterogeneous 3D brain MRI datasets and multiple sequential training orders, establishing CLMU-Net as a strong framework for continual brain lesion segmentation under real clinical variability.

## 2. Methodology

Our goal is to develop a continual brain-lesion segmentation model capable of learning from sequentially arriving datasets, also referred to as episodes or tasks, while remaining robust to shifts in pathology, acquisition sites, and modality availability. In this domain incremental CL setup, the model encounters a sequence of $T$ episodes, $D_1, \ldots, D_T$, each arriving one after another. Our proposed **CLMU-Net** is built around three synergistic components (Fig. 1). First, a modality-flexible input interface accommodates arbitrary and evolving modality sets through dynamic channel inflation and Random Modality Drop (RMD), ensuring generalization and stable performance under heterogeneous imaging protocols. Second, a Domain-Conditioned Textual Guidance (DCTG) module injects global pathology and modality context at the U-Net bottleneck. Each 3D image patch from a patient's image data is paired with a language model derived domain embedding, generated from a prompt describing its lesion type and available modalities, and this embedding interacts with bottleneck features via a cross-attention to produce globally informed, global context-aware representations. Third, a lesion-aware experience replay mechanism maintains long-term stability by storing a balanced mixture of representative and difficult samples from each dataset. After completing $t^{th}$ training session with the current dataset $(D_t)$ and replay buffer $(\mathcal{B}_t^{global})$, each sample from the current dataset $(D_t)$ are ranked using these complementary criteria, and the top-scoring ones are inserted into a fixed-size global buffer, ensuring that future replay batches include both stable anchors of the distribution and challenging cases most susceptible to forgetting. Together, these components enable CLMU-Net to continually acquire new knowledge while preserving past performance, even under shifting modality configurations and non-stationary clinical data streams.

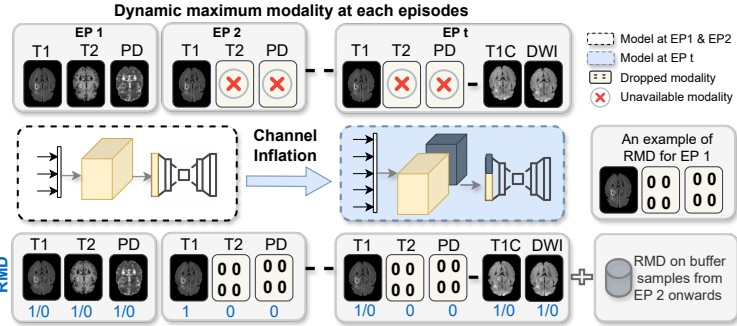

Figure 2: Modality-flexible design: varying episode-wise modalities (top), channel inflation for new modalities (middle), and RMD for modality-agnostic training (bottom).

## 2.1. Buffer Selection Criteria

Experience replay plays a central role in CLMU-Net, and its effectiveness depends critically on which samples are retained in the replay buffer. Randomly selected samples (Rolnick et al., 2019) for buffer can lead to suboptimal or under-utilization of limited buffer capacity. Instead, we employ a lesion-aware selection strategy that balances two complementary categories: *representative* samples that anchor the dominant lesion distribution of past datasets, and *difficult* samples that capture boundary ambiguity and morphological variability. This ensures that the buffer preserves both the stable core of each dataset and the challenging cases most susceptible to forgetting.

**Representative samples:** Representative samples are defined as cases for which the model produces confident lesion predictions and that contain sufficient lesion volume to reflect typical pathology. Such cases anchor the replay buffer to the dominant lesion distribution of each dataset, ensuring adequate coverage of common lesion patterns and supporting stable knowledge retention across datasets. We quantify representativeness using two complementary measures: lesion prediction confidence and lesion size.

For each 3D MRI volume $i$, the network outputs a voxel-wise softmax probability $\hat{p}^{(i)}(v) \in [0,1]$ for the lesion class at each voxel $v$, and the corresponding ground-truth annotation is denoted by $G^{(i)}(v) \in \{0,1\}$. Let $L^{(i)} = \{v \mid G^{(i)}(v) = 1\}$ be the set of lesion voxels and let $\tau = 0.5$ be a confidence threshold. To ensure that the confidence score reflects reliable predictions, only lesion voxels with sufficiently high predicted lesion probability contribute positively. Concretely, we define for each lesion voxel $s^{(i)}(v) = \hat{p}^{(i)}(v)$ if $\hat{p}^{(i)}(v) > \tau$ and $s^{(i)}(v) = 0$ otherwise. The sample-level confidence score is then given by $S_{\text{conf}}^{(i)} = \frac{1}{|L^{(i)}|} \sum_{v \in L^{(i)}} s^{(i)}(v)$. Higher values indicate that a larger fraction of lesion voxels are segmented with high confidence, suggesting that the model has well internalized the lesion appearance for that case. Lesion size provides a second signal of representativeness, as larger lesions offer richer and more diverse supervision. Volumes with larger lesions contribute more positive voxels and better capture the main structure of the pathology, reducing the risk that the buffer is dominated by cases with very small lesions or mostly

background. For each sample $i$, we therefore define the lesion size score as $S_{\text{size}}^{(i)} = |L^{(i)}|$, that is, the number of lesion voxels in the volume. In practice, both scores are normalized across the dataset and combined into a single representativeness score $R_{\text{rep}}^{(i)} = (1-\alpha)\, S_{\text{conf}}^{(i)} + \alpha\, S_{\text{size}}^{(i)}$ using a weighting factor $\alpha \in [0,1]$ that balances the relative importance of prediction confidence and lesion volume when ranking samples for inclusion in the replay buffer.

**Difficult samples:** Difficult cases emphasize boundary ambiguity and irregular morphologies, which are typically the first regions to degrade under distribution shift. Retaining such cases in the buffer ensures that the model remains exposed to challenging lesion structures, thereby improving robustness across sequential tasks. We quantify difficulty using two complementary measures: boundary uncertainty and lesion complexity. Boundary uncertainty characterizes how unstable the model's predictions are near the lesion margin. For each voxel, the network outputs a foreground probability $\hat{p}(v)$, and uncertainty is assessed by evaluating how close these probabilities lie to the decision threshold. Probabilities near this threshold indicate ambiguity, whereas probabilities far from it indicate confident separation of lesion and background. To focus on the most informative region, uncertainty is computed only within a symmetric 3D boundary band of fixed total width of 9 voxels, constructed by expanding the lesion surface by 4 voxels inward and 4 voxels outward. The sample-level boundary uncertainty score is then defined as $S_{\text{unc}}^{(i)} = \frac{1}{|B^{(i)}|} \sum_{v \in B^{(i)}} |\hat{p}(v) - 0.5|$, where $B^{(i)}$ denotes the boundary band for sample $i$. Smaller values indicate greater prediction instability along the lesion margin, making such cases more difficult and more susceptible to forgetting. Further, lesions with fragmented or irregular morphology also pose challenges for sequential learning. We quantify this property using the lesion complexity score $S_{\text{comp}}^{(i)} = (C^{(i)})^2 / N^{(i)}$, where $C^{(i)}$ is the number of connected components and $N^{(i)}$ the number of lesion voxels. Higher values correspond to more irregular or scattered lesions. These two difficulty indicators are combined into a final difficulty score $R_{\text{diff}}^{(i)}$ using a weighting factor $\gamma \in [0,1]$, enabling the sample ranking to reflect both boundary ambiguity and morphological fragmentation.

## 2.2. Final Buffer Composition and Management

CLMU-Net maintains a global replay buffer which is sequentially updated. After completing the $t$-th training session, we compute $R_{\text{rep}}$ and $R_{\text{diff}}$ for all samples in $D_t$ and select the top-ranked volumes in equal proportion from both categories to form the dataset-specific partition $\mathcal{B}_t$. The global buffer is updated by inserting $\mathcal{B}_t$ into the existing buffer that contains partitions from previously seen datasets. Formally, the global buffer after session $t$ is $\mathcal{B}_t^{global} = \mathcal{B}_{t-1}^{global} \cup \mathcal{B}_t$, with $\mathcal{B}_0^{global} = \varnothing$. The buffer has fixed capacity $\beta$, so after insertion we remove samples to satisfy $\sum_{i=1}^{t} |\mathcal{B}_i| = \beta$.

In practice, a simple and effective policy is to maintain approximate parity across partitions by setting $|\mathcal{B}_i| \approx \beta/t$. Importantly, eviction is performed within each partition rather than by comparing ranks across datasets: for any seen partition (i.e., $i \leq t$), we evict samples if it exceeds $\beta/t$. To preserve the fixed balance between representative and difficult samples, eviction is applied category-wise by removing the lowest-ranked samples within each subset of $\mathcal{B}_i$. Consequently, each seen dataset retains a reserved share of the buffer by construction, and a dataset can not be eliminated due to cross-cohort differences in the

scale or distribution of $R_{\text{rep}}$ and $R_{\text{diff}}$. This prevents domination by larger datasets and preserves intra-dataset diversity.

### 2.3. Domain-conditioned Textual Guidance

As illustrated in the DCTG block of Fig. 1, global domain knowledge is injected into the inherently local U-Net bottleneck representation through a text-guided multi-head cross-attention module. For each training case, a short textual description is composed that specifies the lesion type and the set of available MRI modalities (the example in Fig. 1). We adopt a prompt-based representation to avoid assuming a fixed, known-in-advance vocabulary of modalities and lesion descriptors: new modality subsets or lesion types can be expressed in text without redefining the conditioning dimensionality. This text is encoded with a pretrained biomedical language model, BioBERT (Lee et al., 2020), yielding contextual token embeddings $\mathcal{T} \in \mathbb{R}^{B \times N_t \times 768}$.

We keep the text encoder frozen so that the mapping from a given (lesion type, modality subset) description to its embedding is consistent across continual sessions, providing a stable conditioning signal when samples are revisited under replay. The tokens are mapped to the visual feature dimension, resulting in $\tilde{\mathcal{T}} \in \mathbb{R}^{B \times N_t \times d}$ with $d = 256$ and $N_t = 64$. In parallel, the U-Net bottleneck feature map $F \in \mathbb{R}^{B \times C \times H \times W \times D}$ with $C = 256$ is reshaped into a sequence of image tokens $X \in \mathbb{R}^{B \times N_i \times C}$, where $N_i = HWD$. These tokens are linearly projected to $d$ channels, added to a learned positional embedding $P \in \mathbb{R}^{1 \times N_i \times d}$, and normalized, producing $\tilde{X} \in \mathbb{R}^{B \times N_i \times d}$, which serves as the query sequence.

Multi-head cross-attention is then applied with queries derived from the image tokens and keys/values derived from the text tokens: $Q = \tilde{X}W_q$, $K = \tilde{\mathcal{T}}W_k$, and $V = \tilde{\mathcal{T}}W_v$, where each projection is factorized into $h$ heads of dimension $d/h$ ,where $h = 8$. Scaled dot-product attention is computed independently in each head and concatenated, yielding text-conditioned image embeddings $Y \in \mathbb{R}^{B \times N_i \times d}$. A linear projection maps $Y$ back to the original bottleneck channel dimension, after which a residual connection with the original bottleneck tokens and a final layer normalization are applied. The sequence is then reshaped to the original spatial layout, producing the refined bottleneck tensor $F_{\text{DCTG}} \in \mathbb{R}^{B \times C \times H \times W \times D}$. This design enables the bottleneck features to be modulated by cohort-level priors encoded in the textual prompt, such as the expected lesion category and modality configuration, so that the decoder receives locally detailed yet globally informed representations, improving robustness under heterogeneous MRI acquisition protocols.

### 2.4. Modality-flexible Segmentation

Recent works (Xu et al., 2024; Sadegheih et al., 2025b) which facilitate a single U-Net model for multiple brain MRI datasets with heterogenous modality sets assume a fixed maximum number of input channels and represent unavailable modalities with zero-filled placeholders. Although simple, this rigid design limits generalization, since a hospital may acquire a novel modality not considered in this fixed set. A clinically deployable model must accommodate such variability without need to predefine modality layouts. To support arbitrary and evolving modality sets, we equip CLMU-Net with a modality-flexible input layer via channel inflation (Fig. 2, middle). At episode $t$, the input convolution expands to match the maximum number of modalities observed so far, replacing the original $K_{\max}(t-1)$-

channel layer with a $K_{\max}(t)$-channel layer. The computation cost along the CL trajectory is always upper-bounded by a model that fixes its input configuration to $K_{\max}(T)$ from the start. Thus, for any episode where $K_{\max}(t) < K_{\max}(T)$, channel inflation yields marginally lower computation. Newly added channels are zero-initialized, and pretrained weights are copied into the first $K$ channels, allowing seamless continuation of learned representations. Each sample is then mapped to a $K_{\max}(t)$-channel tensor by inserting zero-valued channels for any absent modality.

Further, to improve generalization and reduce spurious correlations between datasets and specific sequences, we adopt RMD (Xu et al., 2024; Sadegheih et al., 2025b) (Fig. 2, bottom). During training, available modalities are randomly masked for both current and replay samples, exposing the model to diverse modality combinations and encouraging redundancy-aware, modality-agnostic features. Together, channel inflation and RMD allow CLMU-Net to operate reliably under arbitrary, incomplete, or newly introduced modality configurations, enabling CL across evolving clinical protocols.

Table 1: Performance comparison (best result, second best result in CL methods).

| | Method (hyperparameter) | S1 | | | S2 | | | Mean | | |
|---|---|---|---|---|---|---|---|---|---|---|
| | | AVG↑ | ILM↑ | BWT↑ | AVG↑ | ILM↑ | BWT↑ | AVG↑ | ILM↑ | BWT↑ |
| UB | Joint | 67.62 | – | – | 67.96 | – | – | 67.79 | – | – |
| UB | Cumulative | 62.37 | 67.40 | -1.60 | 69.20 | 73.04 | 0.05 | 65.78 | 70.22 | -0.78 |
| LB | Naive | 15.73 | 33.64 | -54.14 | 23.43 | 37.36 | -54.16 | 19.58 | 35.50 | -54.15 |
| LB | FromScratchTraining | 16.96 | 31.53 | -57.03 | 14.07 | 26.42 | -24.04 | 15.53 | 28.98 | -40.53 |
| Buffer-free CL | LFL | 18.30 | 30.10 | -53.53 | 11.16 | 31.08 | -60.39 | 14.73 | 30.59 | -56.96 |
| Buffer-free CL | MAS | 37.67 | 50.28 | -4.76 | 34.91 | 47.99 | -6.53 | 36.29 | 49.14 | -5.64 |
| Buffer-free CL | LwF | 29.97 | 41.18 | -45.15 | 18.16 | 36.05 | -57.54 | 24.06 | 38.61 | -51.34 |
| Buffer-free CL | SI | 43.27 | 51.69 | -25.07 | 13.32 | 36.83 | -52.69 | 28.30 | 44.26 | -38.88 |
| Buffer-free CL | EWC | 26.48 | 39.04 | -45.30 | 26.78 | 39.84 | -52.89 | 26.63 | 39.44 | -49.09 |
| Buffer-free CL | MiB | 26.89 | 41.80 | -45.06 | 24.39 | 38.35 | -53.03 | 25.64 | 40.08 | -49.05 |
| Buffer-free CL | TED | 31.49 | 44.08 | -40.86 | 25.76 | 37.63 | -52.26 | 28.62 | 40.86 | -46.56 |
| Buffer-free CL | BrainCL | 54.31 | 56.46 | -16.46 | 32.93 | 51.11 | -27.28 | 43.62 | 53.78 | -21.87 |
| Buffer-based CL | GEM ($\beta$=10) | 45.24 | 54.00 | -24.09 | 36.24 | 48.69 | -34.48 | 40.74 | 51.34 | -29.28 |
| Buffer-based CL | MIR ($\beta$=10) | 19.19 | 36.12 | -51.02 | 19.68 | 35.17 | -55.25 | 19.44 | 35.64 | -53.14 |
| Buffer-based CL | GDumb ($\beta$=10) | 29.42 | 36.14 | -3.74 | 31.57 | 41.85 | -11.00 | 30.50 | 39.00 | -7.37 |
| Buffer-based CL | RCLP ($\beta$=10) | 43.91 | 55.68 | -22.23 | 19.72 | 46.19 | -38.63 | 31.81 | 50.94 | -30.43 |
| Buffer-based CL | ER ($\beta$=10) | 49.56 | 58.57 | -18.18 | 50.11 | 58.65 | -24.12 | 49.84 | 58.61 | -21.15 |
| Buffer-based CL | CLMU-Net ($\beta$=10)+DCTG | 61.25 | 61.89 | -10.44 | 54.22 | 65.67 | -7.59 | 57.73 | 63.78 | -9.02 |
| Buffer-based CL | CLMU-Net ($\beta$=10)+ILI | 63.31 | 63.24 | -11.08 | 54.53 | 66.21 | -9.47 | 58.92 | 64.72 | -10.28 |
| Buffer-based CL | CLMU-Net ($\beta$=10)+ILI+DCTG | 63.15 | 64.40 | -9.83 | 55.30 | 67.66 | -8.63 | 59.22 | 66.03 | -9.23 |

## 3. Experiments

### 3.1. Datasets, Experimental Setup, Evaluation Metrics, CL Benchmarks

**Datasets:** We evaluate and compare CLMU-Net on five heterogeneous 3D brain MRI datasets: BRATS-Decathlon (Bakas et al., 2017), ISLES (Maier et al., 2017), MSSEG (Commowick et al., 2018), ATLAS (Liew et al., 2022), and WMH (Kuijf et al., 2022), covering a wide range of modalities (T1, T1c, T2, PD, DWI, FLAIR), lesion types (tumor, stroke, sclerosis, and white matter hyperintensities), and acquisition centers. Each dataset is treated as a separate episode, arriving sequentially in a domain-incremental CL setting. We tested on two dataset sequences: S1 (BRATS-Decathlon, ATLAS, MSSEG, ISLES, WMH) rep-

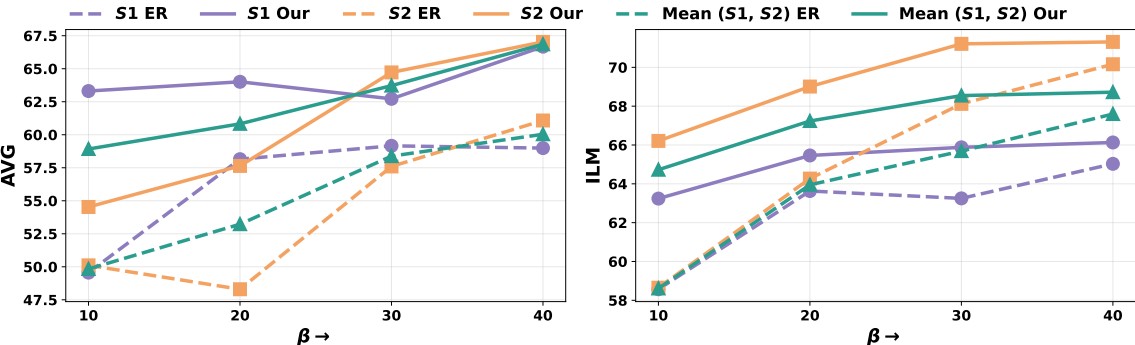

Figure 3: ER (dashed) vs. CLMU-Net (solid) across $\beta$ in $S1$, $S2$ (left/right: AVG/ILM).

resenting large to small dataset sizes and $S2$ (MSSEG, BRATS-Decathlon, ISLES, WMH, ATLAS) representing descending modality counts (Sadegheih et al., 2025b). Train-test split is followed from Sadegheih et al. (2025b).

**Experimental Setup:** All volumes are sampled to a common resolution ($1mm$), skull-stripped, and z-score normalized per modality (Sadegheih et al., 2025b; Xu et al., 2024). During training, we use a patch-wise sampling strategy with size $128^3$ and batch size 2. Optimization is performed using Adam with an initial learning rate of $1 \times 10^{-3}$. Each dataset is trained for 300 epochs before moving to the next training session. We evaluate CLMU-Net and best performing buffer-based method (ER) on different $\beta$ ($\{10, 20, 30, 40\}$). $\alpha, \gamma$ are set as 0.9. All experiments are implemented in PyTorch 2.5 and run on a single NVIDIA H100 GPU with 92 GB memory; training a full task sequence requires approximately 41 GPU hours.

**Evaluation Metrics:** We report the standard volumetric segmentation metric, Dice Similarity Coefficient (DSC), to evaluate model performance across tasks. To evaluate retention across sequential tasks, we adopt popular CL metrics as in previous literature (Kumari et al., 2025b; Sadegheih et al., 2025b): average performance (AVG), incremental learning metric (ILM), backward transfer (BWT) (Lopez-Paz and Ranzato, 2017). Negative BWT reflects forgetting of earlier tasks, while positive values indicate knowledge retention or improvement. The higher the value of these metrics, the higher is the performance.

**CL Benchmarks:** We benchmark CLMU-Net against several representative CL strategies. The lower bound (LB) performance is achieved by 'naive', 'fromScratchTraining' and upper bound (UB) by 'cumulative' and 'joint' methods. We consider frequently benchmarked buffer-free and buffer-based CL methods, also covering those considered in 'Lifelong nnU-Net' (González et al., 2023) (a recent benchmark framework for medical CL). Among buffer-free methods, we consider LFL (Jung et al., 2016), MAS (Aljundi et al., 2018), EWC (Álvarez et al., 2025), SI (Zenke et al., 2017), LwF (Li and Hoiem, 2017), MiB (Cermelli et al., 2020), TED (Zhu et al., 2024), and BrainCL (Sadegheih et al., 2025b). For buffer-based approaches, we consider GEM (Lopez-Paz and Ranzato, 2017), MIR (Aljundi et al., 2019), GDumb (Prabhu et al., 2020), ER (Rolnick et al., 2019), RCLP (Ceccon et al., 2025). We follow the setup presented in BrainCL (Sadegheih et al., 2025b) for re-implementation within the same training pipeline to ensure fair comparison.

### 3.2. Results

**Comparison with other methods:** Table 1 reports the performance of LB, UB, CLMU-Net, and representative CL methods across the five datasets in sequences $S1$, $S2$, and their mean. AVG and ILM summarize segmentation accuracy, while BWT quantifies forgetting. LB and UB are non-CL references included only for contextual comparison. The three metrics must be interpreted jointly. BWT alone can be misleading because it reflects only the change between the current and final session; for example, GDumb shows relatively strong BWT in $S1$ yet yields substantially lower AVG and ILM than other methods. AVG measures only the final-session DSC, whereas ILM captures the mean DSC over all sessions, offering a more complete view of stability. Therefore, conclusions rely on all three metrics.

Buffer-free methods show pronounced degradation, with severe forgetting and substantially lower ILM and AVG values. CLMU-Net clearly outperforms the strongest buffer-free baseline (BrainCL), improving AVG, ILM, BWT by {16.28%, 14.06%, 40.28%} in $S1$ and {67.93%, 32.38%, 68.37%} in $S2$, despite using only ten stored past samples. This highlights the role of replay in mitigating forgetting complex brain lesion segmentation application. Among buffer-based approaches, CLMU-Net achieves the best AVG and ILM and the lowest forgetting. Relative to the strongest baseline in this category (ER), CLMU-Net improves AVG, ILM, BWT by {27.42%, 9.95%, 45.93%} in $S1$ and {10.36%, 15.36%, 64.22%} in $S2$.

When textual guidance (DCTG) and input-layer inflation (ILI) are combined with replay, the hybrid variant outperforms using either component alone, indicating that these modules provide complementary benefits. The top two AVG and ILM scores (red and blue in Table 1) across all CL methods are achieved by CLFU-Net variants, reflecting the strength of this design.

Overall, the lesion-aware buffer strategy in CLMU-Net provides consistent gains across $S1$, $S2$, and their mean, surpassing both buffer-free and buffer-based baselines and demonstrating the advantage of coupling targeted replay with modality-flexible architecture and global textual guidance.

**Comparison with different buffer sizes:** Fig. 3 compares CLMU-Net with the strongest baseline, ER, by reporting AVG and ILM across buffer sizes $\beta$ and sequences $S1$ and $S2$. Both methods improve as $\beta$ increases, yet CLMU-Net consistently surpasses ER, with the largest gains appearing in the low-buffer regime ($\beta \leq 20$). Using the mean performance over $S1$ and $S2$ (green curves in Fig. 3), the relative gains of our method over ER across $\beta \in \{10, 20, 30, 40\}$ are {21.51%, 14.28%, 9.15%, 11.35%} for AVG and {11.50%, 5.14%, 4.35%, 1.66%} for ILM. These results indicate that the lesion-aware selection mechanism captures the underlying data distribution more effectively and provides more informative samples per memory budget, effectively mitigating forgetting even under extremely tight $\beta$.

### 4. Ablation

Table 2 analyzes the contribution of each buffer-selection criterion across $\beta$ and sequences $S1$ and $S2$. Within the difficult samples criterion ($R_{\mathrm{diff}}$), uncertainty (U) outperforms lesion complexity (LC), so the $R_{\mathrm{diff}}$ assigns greater weight to U. Within the representativeness criterion ($R_{\mathrm{rep}}$), lesion size (LS) outperforms corrected confidence (CC), leading to a higher

Table 2: Ablation for buffer selection criterion (U: Uncertainty, LC: Lesion complexity, LS: Lesion Size, CC: Correct confidence).

| Seq. | Buffer criterion | $\beta=10$ | | | $\beta=20$ | | | $\beta=30$ | | | $\beta=40$ | | | Mean over $\beta$ | | |
|---|---|---|---|---|---|---|---|---|---|---|---|---|---|---|---|---|
| | | AVG↑ | ILM↑ | BWT↑ | AVG↑ | ILM↑ | BWT↑ | AVG↑ | ILM↑ | BWT↑ | AVG↑ | ILM↑ | BWT↑ | AVG↑ | ILM↑ | BWT↑ |
| S1 | $R_{diff}$ (U) | 47.22 | 54.87 | -23.22 | 62.12 | 64.46 | -9.76 | 61.25 | 64.68 | -11.17 | 63.94 | 65.87 | -7.52 | 58.63 | 62.47 | -12.92 |
| | $R_{diff}$ (LC) | 44.77 | 54.02 | -25.02 | 49.80 | 57.60 | -19.16 | 58.43 | 61.13 | -12.49 | 54.96 | 60.56 | -13.92 | 51.99 | 58.33 | -17.65 |
| | $R_{rep}$ (LS) | 59.06 | 62.91 | -12.15 | 62.56 | 63.83 | -8.98 | 60.39 | 63.69 | -8.48 | 64.12 | 66.17 | -7.11 | 61.53 | 64.15 | -9.18 |
| | $R_{rep}$ (CC) | 58.35 | 59.79 | -16.65 | 56.32 | 59.84 | -16.0 | 61.61 | 63.1 | -10.37 | 59.28 | 62.84 | -11.67 | 58.89 | 61.39 | -13.67 |
| | **Ours** | **63.31** | **63.24** | **-11.08** | **64.01** | **65.46** | **-8.98** | **62.72** | **65.88** | **-9.48** | **66.67** | **66.13** | **-6.29** | **64.18** | **65.18** | **-8.96** |
| S2 | $R_{diff}$ (U) | 49.67 | 64.98 | -9.80 | 62.53 | 68.8 | -4.79 | 60.31 | 68.89 | -6.32 | 66.24 | 70.61 | -1.89 | 59.69 | 68.32 | -5.70 |
| | $R_{diff}$ (LC) | 52.51 | 64.43 | -12.96 | 57.33 | 68.66 | -8.43 | 58.13 | 68.01 | -7.23 | 57.45 | 68.23 | -6.85 | 56.36 | 67.33 | -8.87 |
| | $R_{rep}$ (LS) | 53.43 | 66.84 | -11.1 | 57.41 | 69.27 | -10.83 | 58.94 | 69.38 | -8.39 | 61.49 | 69.39 | -2.85 | 57.82 | 68.72 | -8.29 |
| | $R_{rep}$ (CC) | 53.51 | 64.23 | -14.98 | 53.15 | 65.13 | -14.59 | 61.02 | 65.97 | -9.85 | 61.48 | 68.91 | -7.96 | 57.29 | 66.06 | -11.85 |
| | **Ours** | **54.53** | **66.21** | **-9.47** | **57.64** | **69.01** | **-7.4** | **64.72** | **71.21** | **-4.82** | **67.03** | **71.31** | **0.34** | **60.98** | **69.44** | **-5.34** |

weight for LS in the $R_{\text{rep}}$. Since U and LS show broadly comparable performance, the buffer $\mathcal{B}_t$ for dataset $D_t$ allocates an equal number of samples from $R_{\text{diff}}$ and $R_{\text{rep}}$.

As $\beta$ increases, $R_{\text{rep}}$ improves stability by anchoring dominant lesion characteristics, while $R_{\text{diff}}$ captures challenging boundary and morphology cases that are more prone to forgetting. When combined, the two criteria yield higher, AVG, ILM, and BWT scores across $\beta$ and sequences. This indicates that jointly capturing both stable lesion structure and difficult boundary cases is necessary for effective replay under heterogeneous sequential streams. The consistent improvements across $\beta$ support that the balanced selection mechanism retains knowledge more effectively than any single criterion.

Compared with the fixed-input design in (Sadegheih et al., 2025b), ILI mechanism in CLMU-Net enables the model to accommodate an arbitrary and previously unknown number of input modalities by dynamically expanding the input layer when new modalities appear, rather than preallocating channels for a maximum set at the start of training. Fixed-input architectures must reserve channels for modalities that are absent in early tasks, leading to under-utilised capacity and potentially suboptimal representations, whereas ILI preserves parameters for previously seen modalities and allocates new channels only when required. As reported in Table 3, this design yields consistent gains over the fixed-input baseline across all values of $\beta$, with average improvements in {AVG, ILM, BWT} of {7.61%, 3.71%, 46.38%} and {9.03%, 3.75%, 41.25%} in $S1$ and $S2$, respectively, indicating better overall segmentation performance and substantially reduced forgetting in sequential multimodal MRI segmentation without prior knowledge of the maximum modality set.

Table 3: Ablation to show impact of input layer inflation (ILI).

| Seq. | ILI | $\beta=10$ | | | $\beta=20$ | | | $\beta=30$ | | | $\beta=40$ | | | Mean over $\beta$ | | |
|---|---|---|---|---|---|---|---|---|---|---|---|---|---|---|---|---|
| | | AVG↑ | ILM↑ | BWT↑ | AVG↑ | ILM↑ | BWT↑ | AVG↑ | ILM↑ | BWT↑ | AVG↑ | ILM↑ | BWT↑ | AVG↑ | ILM↑ | BWT↑ |
| S1 | ✗ | 59.63 | 62.3 | -12.0 | 58.74 | 63.18 | -12.6 | 61.75 | 64.95 | -9.25 | 62.25 | 65.24 | -8.47 | 60.59 | 63.92 | -10.58 |
| | ✓ | **63.31** | **63.24** | **-11.08** | **64.01** | **65.46** | **-8.98** | **62.72** | **65.88** | **-9.48** | **66.67** | **66.13** | **-6.29** | **65.20** | **66.29** | **-5.78** |
| S2 | ✗ | 53.81 | 67.21 | -10.07 | 59.81 | 68.77 | -6.89 | 57.40 | 68.92 | -6.21 | 64.58 | 71.40 | -1.73 | 58.90 | 69.07 | -6.23 |
| | ✓ | **54.53** | **66.21** | **-9.47** | **57.64** | **69.01** | **-7.4** | **64.72** | **71.21** | **-4.82** | **67.03** | **71.31** | **0.34** | **64.22** | **71.66** | **-3.66** |

## 5. Limitations and Future Directions

Our buffer design uses a fixed balance of prototypical and challenging samples for simplicity, yet this ratio may not always yield optimal replay performance, since the relative value of each category can vary across datasets and training stages. More adaptive schemes, such

as dynamically weighting the contribution of prototypical versus challenging samples based on dataset statistics or model behavior, or formulating a unified ranking strategy that integrates both criteria into a single score may yield further gains. Another limitation is the reliance on a stored buffer, which raises privacy concerns in clinical settings where even small storage of raw MRI scans may conflict with governance constraints. This motivates privacy-preserving alternatives such as generative replay, though generating high-fidelity 3D medical volumes remains computationally demanding and data-intensive. Developing adaptive buffer allocation policies and practical generative or prior-driven replay mechanisms represents important direction for future work.

Finally, CLMU-Net is developed around the clinical and technical challenges of continual 3D brain lesion segmentation in multi-modal MRI, including small and spatially dispersed lesions and substantial cross-cohort heterogeneity. While the underlying CL setting (domain shift with variable modality availability) is not inherently brain-specific, evaluating applicability to other organs under comparable variable-modality 3D MRI protocols is a promising direction for future investigation.

## 6. Conclusion

We introduced CLMU-Net, a continual domain incremental framework for brain lesion segmentation that combines lesion-aware replay, modality-flexible processing, and domain-conditioned textual guidance. Unlike prior approaches, CLMU-Net mitigates catastrophic forgetting while accommodating *arbitrary and variable* modality sets. The integration of clinical text enhances global contextual reasoning, improving robustness under heterogeneous pathology and acquisition conditions. Experiments on five diverse 3D brain MRI datasets show consistent gains over state-of-the-art baselines, particularly under strict buffer budgets. These findings demonstrate the effectiveness of lesion-aware replay, flexible modality handling, and domain-aware guidance for continual medical image segmentation.

## Acknowledgments

This work was supported by the German Research Foundation (Deutsche Forschungsgemeinschaft, DFG) under the grant no. 417063796. Further, the authors gratefully acknowledge the computational and data resources provided by the Leibniz Supercomputing Center (www.lrz.de).

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
