# OpenReview forum: "Towards Modality-Agnostic Continual Domain-Incremental Brain Lesion Segmentation"
_MIDL.io/2026/Conference — MIDL 2026 Poster_

### Official Review · Reviewer_XUCH · 2025-12-30

**Confidence:** 3
**Preliminary Rating:** 4
**Final Rating:** 5

**Summary:**

The paper proposes CLMU-Net, a replay-based continual learning (CL) framework for domain-incremental 3D brain lesion segmentation under heterogeneous and changing MRI modality sets. The framework incorporates dynamic input-layer channel inflation mechanism to accommodate unseen modality sets and avoid setting a maximum modality set. Additionally, a lesion-aware replay buffer is proposed that prioritises representative and challenging samples ensuring robustness. Finally, a domain-conditioned text embedding guidance is injected in the network bottleneck to provide global information. The approach is validated on 5 public brain MRI datasets across different sequences and buffer sizes, showing consistent improvement over buffer-free and buffer based CL baselines.

**Strengths:**

Strengths:
- Well-motivated and clinically realistic problem of sequential domain incremental-learning.
- Strong empirical evaluation spanning several datasets and multiple sequences across several benchmarks.
- Robust replay strategy to include both representative and challenging samples in buffer rather than random selection.
- Dynamic input-layer channel inflation to adapt to unseen modalities.
- Meaningful ablation studies examining buffer criteria and role of the inflation strategy.

**Weaknesses:**

Weaknesses:
- Limited analysis of computational cost associated with the proposed channel inflation mechanism, particularly in the 3D replay-based training setting.
- Lack of discussion of practical upper bounds on computation or memory as input channels grow monotonically with the number of encountered modalities.
- Domain-conditioning through crafted sentences and text-encoders is not adequately justified against using simpler information-matched embedding alternatives.
- The provided codebase is empty

**Detailed Comments:**

The author's work presented in the paper is thorough, well-motivated and relevant. To further strengthen the presentation, the authors could more clearly justify the choice of using hand-crafted textual prompts and a BioBERT-based encoder, and clarify why this design was preferred over simpler conditioning mechanisms based on modality availability and lesion labels.

Additionally, while the work is impressive, the authors are encouraged to temper claims of scalability and more explicitly acknowledge the computational and memory implications of monotonic channel growth under the proposed channel inflation mechanism.

**Justification Of Final Rating:**

The authors have adequately addressed the concerns raised in the initial review, and the revised manuscript provides sufficient context to make the presentation more accessible to a non-specialist audience.

**Justification Of The Preliminary Rating:**

The paper presents a technically sound, well-written, and empirically strong contribution to the continual learning literature in medical image analysis. The identified weaknesses and suggestions are intended to strengthen the depth of analysis and to better acknowledge the practical bottlenecks related to computational availability in clinical settings.

**Questions To Address In The Rebuttal:**

N/A

---

> ### Author Response · Authors · 2026-01-24
> **Authors’ Comments (Part I)**
>
> Dear Reviewer XUCH,
> Thank you very much for your detailed and constructive feedback. Your suggestions have significantly helped us to enhance the clarity and robustness of our paper. Below, we provide detailed responses addressing your comments and questions.
> ___
> > Limited analysis of computational cost associated with the proposed channel inflation mechanism, particularly in the 3D replay-based training setting.
> Lack of discussion of practical upper bounds on computation or memory as input channels grow monotonically with the number of encountered modalities.
> The authors are encouraged to temper claims of scalability and more explicitly acknowledge the computational and memory implications of monotonic channel growth under the proposed channel inflation mechanism.
>
> Thank you for the suggestion. We agree that channel inflation (ILI) leads to a monotonic increase in the number of input channels as new modalities are encountered, and we would like to clarify its computational/memory implications more explicitly.
>
> First, the growth is strictly bounded by the maximum MRI sequences across the datasets in our setting. Moreover, ILI affects only the input mapping / first layer, where the channel dimension is defined; the remainder of the 3D U-Net backbone is unchanged. Therefore, both the memory and compute impacts of inflation are limited to a small fraction of the network (the input layer only).
>
> Second, the relevant comparison in continual learning is the trajectory cost. A common approach in previous literature [1,2] is to fix the input layer to the maximum expected MRI modality count from the first episode. ILI avoids this conservative allocation: in early episodes (e.g., starting from ATLAS with a single modality), the model runs with a smaller effective input interface, which yields lower compute/memory than the fixed-max design. As modalities accumulate, the input layer is expanded only when needed.
>
> Finally, by the last episode, both ILI and non-ILI models operate on the same final modality set, so the end-of-training compute is identical; ILI does not increase the computational cost in the last episode beyond what is required by the final modality configuration. Quantitatively, along the ATLAS→…→6-modality trajectory, the total change is minimal: 10.409M → 10.412M parameters (+0.02%) and 137.766G → 138.332G GFLOPs (+0.41%) from the first to the last episode. These results indicate that ILI is cost-neutral at convergence while providing flexibility and efficiency benefits earlier in the continual process. We have added the clarification in the revised manuscript (Section 2.4).
>
> > The provided codebase is empty
>
> Thank you for pointing this out. We have now updated the repository and added the complete codebase required to reproduce the experiments (training, evaluation, and configuration files).
>
> https://github.com/xmindflow/CLMU-Net

---

> > ### Author Response · Authors · 2026-01-24
> > **Authors’ Comments (Part II)**
> >
> > > Domain-conditioning through crafted sentences and text-encoders is not adequately justified against using simpler information-matched embedding alternatives.
> > To further strengthen the presentation, the authors could more clearly justify the choice of using hand-crafted textual prompts and a BioBERT-based encoder, and clarify why this design was preferred over simpler conditioning mechanisms based on modality availability and lesion labels.
> >
> > Thank you for the comment. We agree that the conditioning variables in our setup (lesion type and modality availability) could be encoded with simpler information-matched alternatives (e.g., a multi-hot conditioning vector followed by a small MLP layer). Indeed, in our prior work, we adopted this type of structured conditioning: a binary domain token that concatenates modality and pathology indicators (e.g., $I_{d+m}​$) to facilitate input-aware guidance in a 3D U-Net [1]. However, that work requires prior knowledge about all modalities and lesion counts.
> >
> > In the present work, we intentionally use a template-based textual prompt encoded by a frozen BioBERT because our variable-modality continual-learning setting does not assume that the full set of modalities and lesion descriptors is known a priori. Under this assumption, a fixed-size vector interface must either (i) commit in advance to a closed vocabulary and dimensionality (thereby limiting extensibility), or (ii) introduce configuration-specific embeddings/MLPs or repeated interface redesign as new modality/label configurations appear over time. In contrast, a prompt-based representation provides a configuration-agnostic conditioning interface: new modality subsets or lesion descriptors can be expressed in text while keeping the conditioning dimensionality fixed.
> >
> > We keep BioBERT frozen so that the mapping from a given (lesion type, modality subset) description to its embedding is consistent across continual sessions, and we train only lightweight projection and cross-attention layers to inject this context into the visual feature space. This choice is particularly relevant for replay-based continual learning, where some samples (buffer) are revisited across episodes: a trainable conditioning encoder (e.g., an MLP over structured tokens) can drift over time and assign different embeddings to the same buffered case, introducing additional non-stationarity. We have revised the manuscript (Section 2.3) to make these design choices explicit.
> >
> > # References
> >
> > [1] Yousef Sadegheih, Pratibha Kumari, and Dorit Merhof. Modality-agnostic brain lesion segmentation with privacy-aware continual learning. In International Workshop on PRedictive Intelligence In MEdicine, pages 1–13. Springer, 2025b.
> >
> > [2] Xu, Wentian, et al. "Feasibility and benefits of joint learning from MRI databases with different brain diseases and modalities for segmentation." Medical Imaging with Deep Learning.

---

> ### Comment · Area_Chair_RFAb · 2026-01-30
> **please update your rating**
>
> Hello and thank you again for reviewing for MIDL !
> This is a friendly reminder to please update your rating based on author's rebuttal.
> This is really important to complete the review process and for the acceptance/rejection of papers.
> The deadline is tomorrow (February 1st 2026, 23:59 AoE).
> Thank you!

---

### Official Review · Reviewer_nBCM · 2026-01-09

**Confidence:** 5
**Preliminary Rating:** 4
**Final Rating:** 5

**Summary:**

The work proposes a continual learning method and a machine learning model architecture for optimizing a vision-language model to perform semantic segmentation, under increasing variety of datasets and domains, while minimizing catastrophic forgetting. The paper shows seemingly significant improvements compared to extensive weak and strong baselines.

**Strengths:**

The strengths of the paper revolve around their main contributions:

1) establishing a continual learning method that simply relies on updating a replay buffer, as new datasets arrive, with the best (most representative + most difficult) input-target examples of each new dataset;

2) a simple augmentation scheme that makes use of evolving input layers to accommodate increasing amount of modalities as they appear;

3) a robust ML architecture that combines vision with language by making use of textual information to provide more versatile global and specific context as to what the model is supposed to be observing and what it is tasked to segment.


Combined, these contributions facilitate the development of machine learning models which robustly and consistently support an increasing amount of modalities while minimizing performance degradation. While not novel, the work effectively communicates the effort and brings to the table a new and strongly performing method to tackle such problem.

**Weaknesses:**

The paper has the following weaknesses.

1) The buffer retention method to determine when to 'evict' cases from the replay buffer may be problematic. More specifically, because cases are included into the buffer according to their R factors on a per-dataset basis, after inclusion into the replay buffer the ranking of each case is then compared against the entire buffer without taking into account that different datasets might have different distribution of R factors.
2) The pre-processing methodology details are lacking. Authors mention "skull-stripped", "common resolution", and "z-score normalized". Real-world medical imaging data comes in a large and often inconsistent variety. When dealing with multiple modalities at once, this problem becomes amplified. As such, it is important to clearly describe the steps used during pre-processing, from DICOM (or original source data) to final data passed to the input of the ML model.
3) The order of dataset sequences need to be more extensively validated. With a total of only 41 hours (less than 2 days) per sequence, it is not unreasonable to expect experiments to more deeply verify the impact of the order of training. In some metrics, performance has consistently varied by more than 30% in absolute terms (e.g. -10.44 to -7.59 for CLMU-Net (β=10)+DCTG).

**Detailed Comments:**

The buffer composition method does not take into account properties of the input image. Because the method relies on features of the segmentation masks and/or the probability maps of the models, it only indirectly considers situations in which the input medical volume (CT image, MRI image, etc) and the corresponding target lesion masks are weakly associated. To be more specific, there may be situations in which a large lesion with a clearly defined lesion mask boundary shape could exist for both an input volume which has high tissue differentiation or texture; as well as for an input volume which has little or no such properties. This situation is common in CT images of the brain, for example in the context of Acute Ischemic Stroke where ischemia has not yet yielded significant hypoattenuation in the CT but the lesion is large nonetheless - yet the same lesion size and in the same location in other case may have shown a significant hypoattenuation causing clear visible signs of lesion in the input image.

**Justification Of Final Rating:**

I believe the authors provided grounded justifications for the concerns raised and comments made. The work is promising and I am looking forward to seeing next published work in this area of research.

**Justification Of The Preliminary Rating:**

While some questions need answering to fully digest the conclusions and interpret the statistical significance of the results as a function of varying dataset sequences, the paper is, regardless, a substantial contribution to machine learning in medical imaging, and to the continual learning field of study.

**Questions To Address In The Rebuttal:**

1) Please clarify if buffer retention may exclude full datasets due to all of their cases ranking lower Rrep/Rdiff than other datasets.
2) Please provide more details about the pre-processing pipeline.
3) Please explain how z-scoring was performed and which splits were used exactly for computing the normalizing constant.
4) Please provide a reasonable explanation for the choice of dataset sequences and, time permitting, show results for other sequence variations.

---

> ### Author Response · Authors · 2026-01-24
> **Authors’ Comments (Part I)**
>
> Dear Reviewer nBCM,
> Thank you very much for your detailed and constructive feedback. Your suggestions have significantly helped us to enhance the clarity and robustness of our paper. Below, we provide detailed responses addressing your comments and questions.
>
>
> > The buffer retention method to determine when to 'evict' cases from the replay buffer may be problematic. More specifically, because cases are included into the buffer according to their R factors on a per-dataset basis, after inclusion into the replay buffer the ranking of each case is then compared against the entire buffer without taking into account that different datasets might have different distribution of R factors.
> Please clarify if buffer retention may exclude full datasets due to all of their cases ranking lower Rrep/Rdiff than other datasets.
> The proposed buffer selection is driven by mask/prediction characteristics but ignores image appearance.
>
> Thank you for your concern. We clarify that buffer eviction is not performed by globally ranking samples across datasets. Instead, the replay buffer is maintained as dataset-specific partitions with an approximately equal quota per seen dataset (Sec. 2.2, ∣$\mathcal{B}_i$∣≈β/t). After adding $\mathcal{B}_t​$, capacity is enforced by evicting within partitions that exceed their quota (i.e., separately per dataset), and category-wise to preserve the fixed representative/difficult split (removing the lowest-ranked samples within each subset of $\mathcal{B}_i​$). Consequently, no dataset can be entirely excluded due to cross-dataset differences in the scale/distribution of R_rep or R_diff. An example is provided below for clarification. We have made this eviction rule explicit in Section 2.2 of the revised manuscript.
>
> For example, if β=40,
>
> - After training with episode 1: the buffer is filled with D1:40 samples
>
> - After training with episode 2: the buffer is filled with D1:20 (20 were evicted to make room for D2) and D2:20 samples
>
> …(and so on)..
>
> - After training with episode 4: the buffer is filled with D1:10, D2:10 samples, D3:10 samples, D4:10 samples
>
>
> Regarding your insightful observation about the impact of input image intensities:
>
> Our buffer selection is lesion- and model-centric: it uses lesion statistics (e.g., size/complexity) together with prediction-derived signals (e.g., confidence and boundary uncertainty from the probability map), rather than explicit handcrafted appearance cues. In cases where the lesion is large but visually subtle (weak image-mask association), the model often shows reduced confidence and/or elevated uncertainty, increasing the likelihood that the sample is retained by the “challenging” criterion (R_diff). Conversely, visually easier and more typical samples tend to receive higher confidence and are preferentially covered by the “representativeness” criterion (R_rep​).
>
> While we do not explicitly compute hand-crafted texture/contrast descriptors, the “challenging” criterion of our buffer selection is driven by the model’s predictive uncertainty, which is computed from the input volume and therefore serves as an implicit proxy for image-level difficulty (e.g., low-contrast lesions or ambiguous tissue differentiation typically yield higher uncertainty).
> We agree that adding explicit appearance-based criteria (contrast/edge strength or feature-space diversity) is an interesting direction. However,  in a 3D continual setting with variable modality availability, it would introduce additional computation and per-volume analysis/feature extraction, and would require careful modality-specific design strategies; we thus leave this for future investigation.

---

> > ### Author Response · Authors · 2026-01-24
> > **Authors’ Comments (Part II)**
> >
> > > The pre-processing methodology details are lacking. Authors mention "skull-stripped", "common resolution", and "z-score normalized". Real-world medical imaging data comes in a large and often inconsistent variety. When dealing with multiple modalities at once, this problem becomes amplified. As such, it is important to clearly describe the steps used during pre-processing, from DICOM (or original source data) to final data passed to the input of the ML model.
> > Please provide more details about the pre-processing pipeline.
> > Please explain how z-scoring was performed and which splits were used exactly for computing the normalizing constant.
> >
> > Thank you for highlighting the need for more detailed description of the pre-processing pipeline. We agree that, especially in multi-cohort and multi-modal MRI settings, pre-processing choices can substantially affect both performance and comparability. In the revised manuscript, we have now added the citations of the publications that were followed for the preprocessing [1,2].
> >
> > **Data format and initial handling**: All datasets used in this study are publicly released in NIfTI format (or provided with an accompanying NIfTI conversion by the dataset organizers). Therefore, our pipeline starts from the released NIfTI volumes rather than raw DICOM.
> >
> > **Pre-processing pipeline (per subject, per modality)**: For each subject and each available MRI modality, we apply the following steps before forming the multi-modal input:
> >
> > - Skull stripping / brain masking: We apply skull stripping to remove non-brain tissues and reduce inter-dataset variability introduced by differences in head coverage and background structure. The resulting brain mask is also used to define “foreground voxels” for intensity normalization.
> > - Resampling to a common voxel spacing (“common resolution”): Each modality volume (and the corresponding segmentation label map) is resampled to a fixed target spacing to ensure consistency across cohorts acquired with different voxel sizes. Image volumes are resampled with continuous interpolation, while label maps are resampled with nearest-neighbor interpolation to preserve discrete class boundaries.
> > - Intensity normalization (z-score): We use per-volume, per-modality z-score normalization computed on foreground voxels only: $x′=(x−μ_{fg})​/​​σ_{fg}$. where $μ_{fg}$​ and $σ_{fg}$ are computed from voxels inside the brain mask (i.e., excluding background). This is performed independently for each subject and each modality, which is a common practice for MRI to handle scanner- and protocol-dependent intensity scaling [3].
> >
> > **Clarification on splits and “normalizing constants”**: Importantly, our z-scoring does not rely on dataset-level statistics (no global mean/std computed across training subjects). Instead, $μ_{fg}$​ and $σ_{fg}$​ are computed within each individual volume. Consequently, there is no leakage risk across train/test splits for normalization constants, because no information is aggregated across subjects or across splits.
> >
> > >  The order of dataset sequences need to be more extensively validated. With a total of only 41 hours (less than 2 days) per sequence, it is not unreasonable to expect experiments to more deeply verify the impact of the order of training. In some metrics, performance has consistently varied by more than 30% in absolute terms (e.g. -10.44 to -7.59 for CLMU-Net (β=10)+DCTG).
> > Please provide a reasonable explanation for the choice of dataset sequences and, time permitting, show results for other sequence variations.
> >
> > Thank you for highlighting the importance of dataset order in domain-incremental continual segmentation. We fully agree that the training sequence can materially affect CL metrics, and that the variability you point out reflects a broader and well-known characteristic of continual learning which is order-induced interference.
> >
> > We adopted the two sequences used in the recent state-of-the-art work we benchmark against [1] to ensure direct and fair comparability under an identical CL protocol. Concretely, S1 (BRATS→ATLAS→MSSEG→ISLES→WMH) follows a large-to-small dataset-size ordering, while S2 (MSSEG→BRATS→ISLES→WMH→ATLAS) follows a descending modality-count ordering. Hence, we followed these two sequences that were extensively analyzed (main results and ablations) in [1].
> >
> > **(Response continues in the next comment section.)**

---

> > ### Author Response · Authors · 2026-01-24
> > **Authors’ Comments (Part III)**
> >
> > In our 3D multi-modal setup, a full CL training run for a single method on one sequence takes ~41 GPU-hours (H100, 92GB). Expanding sequence coverage “consistently with our paper” would require re-running not only the main benchmark table, but also the buffer-size sweeps and the ablation studies. Even adding one additional sequence across the full experimental grid would therefore require re-running the complete set of experiments (=46 runs), corresponding to 46×41 = 1,886 GPU-hours for that single sequence, which is not feasible within the rebuttal time and allocated compute budget.
> >
> > Despite these constraints, we initiated runs in the rebuttal time window for two additional sequences to address the reviewer’s concern: S3: ISLES→WMH→MSSEG→BRATS→ATLAS (low→high dataset size) and S4: ATLAS→WMH→ISLES→BRATS→MSSEG (ascending modality count)
> >
> > | Method | S3 AVG | S3 ILM | S3 BWT | S4 AVG | S4 ILM | S4 BWT | Mean AVG | Mean ILM | Mean BWT |
> > |---|---:|---:|---:|---:|---:|---:|---:|---:|---:|
> > | Joint | 66.93 | - | - | 64.36 | - | - | 65.65 | - | - |
> > | Cumulative | 67.12 | 67.54 | -0.09 | 66.79 | 61.04 | -1.33 | 66.96 | 64.29 | -0.71 |
> > | Naive | 24.78 | 39.49 | -41.01 | 36.66 | 39.58 | -38.69 | 30.72 | 39.53 | -39.85 |
> > | LwF | 24.82 | 38.24 | -40.10 | 39.71 | 39.68 | -38.10 | 32.27 | 38.96 | -39.10 |
> > | SI | 31.57 | 41.16 | -37.79 | 47.06 | 43.87 | -25.94 | 39.31 | 42.52 | -31.87 |
> > | EWC | 34.19 | 41.55 | -37.65 | 39.93 | 40.06 | -34.97 | 37.06 | 40.80 | -36.31 |
> > | MiB | 31.28 | 39.37 | -39.83 | 39.47 | 39.43 | -35.96 | 35.38 | 39.40 | -37.89 |
> > | TED | 33.44 | 38.46 | -28.91 | 42.08 | 43.69 | -32.92 | 37.76 | 41.08 | -30.91 |
> > | BrainCL | 35.85 | 46.13 | -21.09 | 50.67 | 48.54 | -21.37 | 43.26 | 47.34 | -21.23 |
> > | ER (β=10) | 41.22 | 48.05 | -33.88 | 48.38 | 45.67 | -30.07 | 44.80 | 46.86 | -31.98 |
> > | CLMU-Net (β=10)+DCTG | 51.88 | 58.04 | -20.69 | 51.4 | 45.68 | -31.93 | 51.64 | 51.86 | -26.31 |
> > | CLMU-Net (β=10)+ILI | 50.45 | 57.56 | -19.89 | 50.71 | 47.21 | -28.46 | 50.58 | 52.39 | -24.18 |
> > | CLMU-Net (β=10)+ILI+DCTG | 52.21 | 58.27 | -20.91 | 54.42 | 47.2 | -24.19 | 53.31 | 52.73 | -22.55 |
> >
> >
> > The results reported in the table above align with the conclusions drawn from sequences S1 and S2. In particular, while absolute performance varies with dataset order (as expected in continual learning), the proposed CLMU-Net variants remain consistently top-performing across both additional sequences. Across S3 and S4, CLMU-Net achieves the best overall continual-learning trade-off among all baselines and maintains the top rank when averaging over sequences. For instance, CLMU-Net (β=10)+ILI+DCTG attains the strongest mean performance over S3/S4 (Mean AVG = 53.31, Mean BWT = −22.55), remaining clearly above ER and all buffer-free methods. These results support our central claim that the proposed components (lesion-aware replay, modality-flexible design, and domain guidance) improve robustness to sequence-induced variability rather than benefiting from a particular dataset ordering.
> >
> > Finally, to make the manuscript clearer and more self-contained, we have added explicit justification of S1/S2 selection (comparability to [1] and principled ordering criteria).
> >
> > # References
> >
> > [1] Yousef Sadegheih, Pratibha Kumari, and Dorit Merhof. Modality-agnostic brain lesion segmentation with privacy-aware continual learning. In International Workshop on PRedictive Intelligence In MEdicine, pages 1–13. Springer, 2025b.
> >
> > [2] Xu, Wentian, et al. "Feasibility and benefits of joint learning from MRI databases with different brain diseases and modalities for segmentation." Medical Imaging with Deep Learning.
> >
> > [3] Isensee, Fabian, et al. "nnU-Net: a self-configuring method for deep learning-based biomedical image segmentation." Nature methods 18.2 (2021): 203-211.

---

> ### Comment · Area_Chair_RFAb · 2026-01-30
> **please update your rating**
>
> Hello and thank you again for reviewing for MIDL !
> This is a friendly reminder to please update your rating based on author's rebuttal.
> This is really important to complete the review process and for the acceptance/rejection of papers.
> The deadline is tomorrow (February 1st 2026, 23:59 AoE).
> Thank you!

---

### Official Review · Reviewer_7xrF · 2026-01-10

**Confidence:** 3
**Preliminary Rating:** 2
**Final Rating:** 3

**Summary:**

This paper proposes a U-Net based framework for 3D brain lesion segmentation. It incorporates a lightweight domain-conditioned textual embeddings that provide a global modality-disease context. The experiments show improvement over other baseline models. The authors evaluated the proposed CLMU-Net on five diverse brain MRI datasets.

**Strengths:**

1. Evaluates CLMU-Net on five diverse brain MRI datasets.
2. The proposed method outperforms both buffer-free and rehearsal-based baselines.
3. Develops a domain-conditioned textual guidance mechanism.

**Weaknesses:**

1. This paper didn't compare their segmentation performance with nnUnet which is mostly used as a baseline.
2. The generalizability of this method is questionable.
3. The technical novelty is limited.

**Detailed Comments:**

1. For medical image segmentation, nnUnet is considered one of the most popular baselines because it tends to get good performance over any medical data. However, this paper doesn't include this nnUnet model as a baseline, which questions the credibility of the proposed model in performance.

2. This is only applicable for Brain images. It would be better to see if this can be extended for other applications.

**Justification Of Final Rating:**

The authors have clarified their contribution and the scope of their work. Moreover, they provided a clear outline of their technical novelty, which is the proposed unified framework with three coupled components.
I am increasing my rating by 1 for their added justification.

**Justification Of The Preliminary Rating:**

As the paper doesn't compare their model's performance with a strong baseline like nnUnet, the contribution is questionable. I would definitely change my rating if I see improvement over the nnUnet model because this model is a very strong baseline and is used as a standard model.

**Questions To Address In The Rebuttal:**

Please check the detailed comments and weaknesses.

---

> ### Author Response · Authors · 2026-01-24
> **Authors’ Comments (Part I)**
>
> Dear Reviewer 7xrF,
>
> Thank you very much for your detailed and constructive feedback. Your suggestions have significantly helped us to enhance the clarity and robustness of our paper. Below, we provide detailed responses addressing your comments and questions.
>
> > This paper didn't compare their segmentation performance with nnUnet which is mostly used as a baseline.
> For medical image segmentation, nnUnet is considered one of the most popular baselines because it tends to get good performance over any medical data. However, this paper doesn't include this nnUnet model as a baseline, which questions the credibility of the proposed model in performance.
>
> Thank you for raising the important point about nnU-Net. We agree that nnU-Net is one of the most popular and competitive baselines for single-dataset, train-from-scratch medical image segmentation, and  it is often expected in conventional segmentation benchmarks.
>
> However, our work targets a different problem: domain-incremental continual medical image segmentation, where multiple heterogeneous datasets arrive sequentially (with changing pathology, acquisition protocols, and variable modality availability) and the goal is to continually update one model while minimizing catastrophic forgetting (CF). In this setting, including “standard nnU-Net” as a baseline is not a straightforward or necessarily fair comparison for three reasons:
>
> - nnU-Net is not a single fixed model; it auto-configures differently per dataset.
> A defining feature of nnU-Net is automated adaptation of preprocessing and network/training configuration to each dataset [1]. As a result, across different datasets it may select different patch sizes, input resolutions, depth/capacity, and training schedules and typically assumes a fixed input modality configuration for the given dataset. In our CL setting, where domains arrive sequentially and modality sets can change, “nnU-Net” is no longer a uniquely defined baseline: one must decide which configuration to use at each episode, how to handle changing input channels, and whether/when to reconfigure. These choices materially influence performance and thus introduce additional degrees of freedom unrelated to the continual-learning strategy itself.
> - Re-running nnU-Net from scratch per episode conflicts with the continual-learning objective.
> nnU-Net’s strongest use case is training from scratch per dataset to maximize within-domain performance [1]. In continual learning, repeatedly training from scratch at each episode discards previously acquired knowledge and does not produce a single continually updated model, which is central to our development. For this reason, we already include FromScratchTraining as a lower-bound reference that captures the “train-from-scratch per domain” philosophy under the same controlled pipeline used for all methods (Table 1). We also include Naive sequential fine-tuning as a lower bound for the “continue training” regime (i.e., loading the previous weights and training on the new incoming dataset without any CL-specific components), providing a fair reference for how much forgetting occurs when no CF reduction mechanism is used.
> - The goal is to compare continual-learning strategies, not frameworks or backbones, consistent with prior CL segmentation literature.
> In continual learning studies, it is standard to keep the training/evaluation pipeline fixed and compare the isolated CL mechanism (e.g., replay/distillation/regularization), rather than changing the entire framework, preprocessing, or auto-configuration. For example, Zhu et al. [5] propose a CL strategy (tri-enhanced distillation) and assess it in a controlled experimental setup to quantify the effect of their proposed CL mechanism compared with other state-of-the-art CL strategies, while keeping the underlying segmentation pipeline consistent.
> Similarly, Lifelong nnU-Net [2] is not a drop-in “nnU-Net baseline,” but a substantial adaptation of the nnU-Net framework to support continual learning; they modify the framework and re-implement CL strategies within that adapted environment to enable comparisons across different CL methods. This further underscores the key point: once nnU-Net is adapted for continual learning, the comparison becomes “a particular CL-adapted nnU-Net variant”, not the standard nnU-Net baseline used in single-dataset segmentation.
>
> **(Response continues in the next comment section.)**

---

> > ### Author Response · Authors · 2026-01-24
> > **Authors’ Comments (Part II)**
> >
> > For these reasons, we intentionally keep the segmentation framework fixed and evaluate continual-learning mechanisms in a controlled and reproducible setting. Leveraging established CL libraries (Mammoth [3], Avalanche [4]), we compare against a broad set of strong and recent CL baselines, covering both buffer-free and replay-based methods (e.g., ER [6], TED [5], and RCLP[7]), and include clear upper (Joint/Cumulative) and lower (Naive/FromScratchTraining) reference points. This isolates the contribution of our continual-learning strategy (modality-flexible design and lesion-aware replay) from confounding differences in backbone, preprocessing, or framework configuration.
> >
> > We hope the above discussion clarifies that our evaluation is designed to be methodologically consistent and fair for CL strategies, and that nnU-Net is best viewed as a strong non-continual train-from-scratch reference rather than a direct continual-learning baseline.
> >
> > > The generalizability of this method is questionable. This is only applicable for Brain images. It would be better to see if this can be extended for other applications.
> >
> > We thank the reviewer for his insightful comment. We would like to clarify that our work and contributions are scoped to domain-incremental 3D brain lesion segmentation under variable MRI modality availability, and we therefore do not claim cross-organ generalization in this paper. Our contribution is a continual-learning framework tailored to settings where (i) cohorts from different sources arrive sequentially (domain shift) and (ii) the available modality subset changes over time.
> > An evaluation on other organs would require multi-cohort 3D MRI segmentation benchmarks that simultaneously exhibit domain shift and heterogeneous/variable modality sets, thereby enabling fair and direct comparable assessments; such public 3D datasets are currently far less standardized outside neuro-MRI. While our method and experiments are brain-specific by design, the proposed pipeline is modular and can be adapted by readers to other applications that satisfy the same criteria (sequential domains + heterogeneous/variable modalities) to enable a fair continual evaluation.
> > However, we appreciate the suggestion to explore other organs. We have explicitly noted this as a future direction in the revised manuscript (Section 5).
> >
> > > The technical novelty is limited.
> >
> > Thank you for the comment. We believe the technical novelty is substantive and is summarized in the final paragraph of Section 1: we study domain-incremental 3D lesion segmentation under open-set modality evolution, i.e., sequential cohorts with shifting distributions while the available modality subset changes over time without assuming a fixed, known-in-advance modality set.
> >
> >
> > To address this setting, we introduce a unified framework with three coupled components: (i) Incremental Layer Inflation (ILI) to expand the input interface as new modalities appear while supporting arbitrary observed subsets; (ii) a Domain-Conditioned Textual Guidance (DCTG) module that injects BioBERT-encoded modality/lesion descriptors into the U-Net bottleneck via cross-attention to provide global context to inherently local bottleneck features, yielding fixed-dimensional guidance compatible with evolving modality subsets and without assuming a predefined modality/label vocabulary; (Additional discussion of this design choice is provided in our response to Reviewer XUCH, Comment 3.); and (iii) a lesion-aware replay buffer that selects representative and difficult cases under a low memory budget using segmentation-relevant signals. These design choices are supported by ablations and comparisons (Table 1), where adding our replay, ILI, and DCTG components improves mean CL performance (e.g., ILM 63.78 →64.72→ 66.03) and the full method outperforms strong baselines such as ER (58.61) with less severe forgetting (BWT) across different buffer sizes. Importantly, even with a very small buffer sizes (β=10/20), our proposed buffer selection design which prioritizes both representative and difficult cases exhibit large gains over ER (mean ILM over S1 and S2: 64.72/67.24 vs. 58.61/63.95 for ER).
> >
> > **(Response continues in the next comment section.)**

---

> > > ### Author Response · Authors · 2026-01-24
> > > **Authors’ Comments (Part III)**
> > >
> > > To the best of our knowledge, this is the first work for continual 3D brain MRI lesion segmentation that does not consider maximum modality set a priori; it explicitly supports modality evolution and provides a mechanism to continually update the segmentation model with shifting dataset sources. Notably, our results narrow the gap to the upper-bound cumulative training: averaged over sequences S1 and S2, the mean ILM is 70.22 for cumulative, 66.03 for our method, compared to 58.61 for ER and 53.78 for BrainCL. We believe this advances continual 3D medical image segmentation toward more realistic lifelong deployment, where acquisition protocols and available modalities evolve over time, and it provides a concrete foundation for future modality-agnostic continual segmentation benchmarks and methods. Additionally, our open-source code availability would benefit the community for building and evaluating CL-based segmentation methods under open-set modality availability, which is also relevant to other clinical applications facing heterogeneous and changing imaging protocols.
> > >
> > >
> > > # References
> > >
> > > [1] https://github.com/MIC-DKFZ/nnUNet?tab=readme-ov-file#where-does-nnu-net-perform-well-and-where-does-it-not-perform
> > >
> > > [2] González, C., Ranem, A., Pinto dos Santos, D., Othman, A., & Mukhopadhyay, A. (2023). Lifelong nnu-net: a framework for standardized medical continual learning. Scientific Reports, 13(1), 9381.
> > >
> > > [3] https://github.com/aimagelab/mammoth
> > >
> > > [4] Lomonaco, V., Pellegrini, L., Cossu, A., Carta, A., Graffieti, G., Hayes, T. L., ... & Maltoni, D. (2021). Avalanche: an end-to-end library for continual learning. In Proceedings of the IEEE/CVF conference on computer vision and pattern recognition (pp. 3600-3610).
> > >
> > > [5] Zhu, Zhanshi, et al. "Boosting knowledge diversity, accuracy, and stability via tri-enhanced distillation for domain continual medical image segmentation." Medical image analysis 94 (2024): 103112.
> > >
> > > [6] Rolnick, David, et al. "Experience replay for continual learning." Advances in neural information processing systems 32 (2019).
> > >
> > > [7] Ceccon, Marina, et al. "Multi-label continual learning for the medical domain: A novel benchmark." 2025 IEEE/CVF Winter Conference on Applications of Computer Vision (WACV). IEEE, 2025.

---

> ### Comment · Area_Chair_RFAb · 2026-01-30
> **Please update your rating**
>
> Hello and thank you again for reviewing for MIDL !
> This is a friendly reminder to please update your rating based on author's rebuttal.
> This is really important to complete the review process and for the acceptance/rejection of papers.
> The deadline is tomorrow (February 1st 2026, 23:59 AoE).
> Thank you!

---

### Author Rebuttal · Authors · 2026-01-24

**Rebuttal:**

We have thoroughly revised the manuscript to address the reviewers' concerns and incorporate their suggestions. Changes are highlighted in yellow color in the updated manuscript. Point-to-point responses to the reviewers’ comments are provided under each Reviewer’s session.

Here is a summary of major changes and experiments carried out in the rebuttal phase.

- Pre-processing pipeline: We clarified in the revised manuscript that our preprocessing follows established protocols reported in prior work, and we now reference those sources explicitly.
- Buffer management clarification: We revised Sec. 2.2 to explicitly describe the dataset-partitioned buffer quotas (approximately (∣$\mathcal{B}_i$∣≈β/t) and the within-partition eviction rule (including representative/difficult subsets), clarifying that each seen dataset remains represented in the replay buffer.
- Domain-conditioning justification: Added methodological rationale for text-based domain encoding using a frozen BioBERT encoder, emphasizing why this design is suitable under variable modality availability in CL.
- Reproducibility: Released the full codebase publicly: https://github.com/xmindflow/CLMU-Net
- Channel inflation cost analysis: Clarified the compute and memory implications of channel inflation, with explicit discussion of the associated resource requirements.
- Scope and future work: Explicitly noted extension to other organs as a future direction to better delineate the scope of the current evaluation.
- We clarified the choice of sequences S1 and S2 in the manuscript.

We deeply appreciate the reviewers' positive feedback and recognition of our work's value.

**Supporting Material:**

/attachment/c137dc04a8a9007d02182b34b9e98555bda4b7de.zip

---

### Meta-Review · Area_Chair_RFAb · 2026-02-06

**Recommendation:** Accept (Poster)
**Confidence:** 4

**Metareview:**

The reviewers have appreciated a well-motivated and sound method, with the flexible number of input channels and textual-guidance mechanisms employed for this continual learning paper. They also found the experiments convincing, where the presented method outperforms baselines in a variety of brain lesion datasets.

The authors have also satisfyingly responded to concerns about the selection of buffer examples, generalizability, computational costs, and text prompts.

However, I would like to temper the relatively good ratings (5,5,3) in the light of elements not raised during the reviews.
- First, this method assumes simultaneous access to several datasets through the buffer mechanism. This hypothesis is not realistic, especially in the clinic, where datasets are only accessible one at a time (this motivated the very popular field of source-free domain-adaptation). This point significantly reduces the feasibility/applications of the proposed method.
- Then, the method assumes that every scan comes with a known modality and diagnosis. This, again, is not realistic, especially in the clinic, where the modality of each scan is often not written by practicians (or lost), and diagnosis about tumour/lesion/stroke/hyper-intensity can sometimes be known long after the scan. This point further decreases the clinical applicability of this work.
- Finally, I still think it would have been interesting to evaluate the proposed method against nnUnet, despite not being a continual-leaning pipeline. I emphasise that continual-learning is not an objective in itself, it’s the accuracy of lesion segmentation that matters. In this paper, CL is motivated by the fact that it represents an improvement over traditional modality-specific networks (see intro), because it has access to more varied data and can build robust representations using a flexible number of inputs. Thus, evaluating a nnUnet could have helped testing the reliability of the CL framework, which is at the core of this paper, since CL is presented as one of the main contributions.

To conclude, this paper is a tough case, where reviews missed several important points. I still recommend acceptance because this paper has merits, but only by a thin margin.

---

### Decision · Program_Chairs · 2026-02-13

Accept (Poster)